# Visualizing recovery progress for patients after a stroke

**ABSTRACT**

Stroke is one of the leading causes of disability worldwide. The efficacy of stroke recovery is determined by various factors, including patient adherence to their rehabilitation program. One way to increase patient adherence to their rehabilitation program is to involve patients in their care by visually showing them their recovery progress. Aiming to design visualizations that could best represent stroke recovery, we (1) conducted semi-structured interviews with healthcare providers with expertise with inpatient stroke recovery. Based on design requirements and themes extracted from the interviews, we (2) designed medium-fidelity visualization prototypes representing stroke recovery. Last, we (3) sought feedback on the visualization designs from healthcare providers and applied their recommendations. By providing an integrated overview for both patients and healthcare providers, our visualization can reduce the burden of understanding all aspects of patients' health progress after a stroke that are currently presented separately in the electronic medical record system. Additionally, our visualization could support patients experiencing cognitive and linguistic deficits after a stroke to better understand their rehabilitation progress.

## 1 INTRODUCTION

More than 89,000 strokes occur each year in Canada, and this number is predicted to affect an increasing number of individuals due to population growth and aging [31]. Stroke is a cardiovascular disease that occurs when blood flow to the brain is stopped or diminished, preventing a portion of the brain tissue from receiving oxygen. The functions linked to this area of the brain are disrupted, which may result in neuromotor or cognitive dysfunction [48].

Strokes are debilitating conditions that can require physical and cognitive rehabilitation for months to recover and are a significant source of stress for patients and their families. Caring for patients after a stroke is costly to the healthcare system; these patients are often treated by an interdisciplinary team that designs recovery programs to improve motor function, postural control, and mobility. Adherence to these programs is the key to recovery [15]. However, an extended stay in a rehabilitation center can be tiring and frustrating for patients, and a lack of motivation for goal-directed activities can reduce engagement and benefits from rehabilitation [51] and impending stroke recovery. Tracking and reviewing recovery progress can provide tangible feedback to motivate patients and reinforce their adherence to rehabilitation programs.

The complex and various recovery progress data collected over weeks or months can be demanding and challenging for patients and healthcare providers. Data is simply a collection of raw information that is not always comprehensible. Visualizing data is one of the most effective methods for providing insight and facilitating data-driven decision-making [13]. An effective data visualization can encompass all necessary information while being simple enough to adequately convey that information to the user. Visualizing health data can accurately show a summary of the data in an intuitive, simple, and accessible way [19] to improve patients' comprehension of their health status, increase engagement in care, and encourage the adoption of positive health behaviours [50].

The purpose of this study was to design visualizations representing the rehabilitation progress of patients recovering from a stroke. We gathered preliminary information through semi-structured interviews with healthcare providers from different specialties with at least one year of experience caring for patients recovering from a stroke at an inpatient rehabilitation facility. Based on the themes and design requirements extracted from the interviews, we designed sketches and medium-fidelity prototypes to visualize health progress in patients recovering from a stroke. Lastly, we conducted exit interviews with the same team of healthcare providers to reflect on our designs.

We found that our visualization designs have the potential to enhance both patients' and healthcare providers' experience reviewing recovery progress after a stroke by providing an integrated overview of a collection of health assessments. Particularly, our designs have the potential to increase comprehension of rehabilitation progress in patients with cognitive or linguistic deficits commonly experienced after a stroke and improve communication between patients and healthcare providers.

Our contributions to this paper are as follows:
1. Identified the content, procedure, and technological needs of communicating recovery progress between patients after a stroke and their healthcare provider team.
2. Presented medium-fidelity data visualization designs representing stroke recovery progress.
3. Discussed the potential impacts of visualizing recovery after a stroke in patients with cognitive and linguistic deficits.

## 2 RELATED WORK

In this section, we provide a brief overview of the importance and application of medical data visualizations for healthcare providers and patients. Then, we outline relevant studies on patient-facing visualizations that have assisted in recovery for both in-clinic and at-home rehabilitation, including stroke recovery.

### 2.1 Medical data visualizations

Visualizing medical data is a technique for organizing large amounts of data to extract and present valuable information [53]. Visualizing medical data has gained widespread interest because of its usefulness for patients and healthcare providers in interpreting data analytics faster, recognizing trends, making better decisions, and engaging and informing patients about their care [36, 43]. Medical data is tracked from many different sources, including Electronic Health Records (EHR), Electronic Medical Records (EMR), remote monitoring devices, diagnostic centers, laboratories, pharmaceutical companies, and many Internet of Things (IoT) devices within and outside the hospital.

Visual overviews of clinical patient data have primarily focused on displaying large amounts of patient health data, particularly EHR. EHR data are typically large, diverse, and temporal, which makes them difficult to understand. EHR data visualization and visual analytics allow healthcare providers and patients to better explore, understand, and make choices about patient's health. Visualization of overviews and summaries of clinical patient data has been well studied. For example, Lifelines [33] is an interface that provides a visual overview and facilitates the navigation and analysis of clinical patient records. Outflow [56] is an interactive visualization that summarizes temporal event data extracted from medical data and can be used to analyze congestive heart failure progression

pathways and their outcomes. Visualizing medical data has also been used in healthcare to advance medication safety [54], intensive care patient management [16], patient wellness [20], implementation of healthcare guidelines [8], and to help patients make decisions about their healthcare management strategies [14].

Research has shown that patients in the hospital want to be educated about their care but often lack the resources to fully interact with their health information and treatment [22]. The inability to obtain this information affects patients' capacity to participate in the treatment they receive [5]. Given the high number of hospitalizations for stroke occurring in Canada each year, it is necessary to harness the benefits of interactive data visualization to assist healthcare providers while informing and engaging recovering patients. Interactive data visualizations representing stroke recovery progress could provide healthcare providers with a visual overview to facilitate communication with patients, and increase stroke survivors' engagement in their health management and understanding of their health status.

## 2.2 Visualizing health recovery data

Visualization of medical data has been widely applied to aid in various forms of rehabilitation. This section introduces visualization tools that represent medical data for patients recovering in clinics and home environments.

### 2.2.1 Visualizing recovery at home

As outpatient treatments are becoming increasingly popular in the healthcare industry, rapid advancements in technology have played a crucial role in creating new tools that allow healthcare providers to monitor, treat, and educate patients recovering at home while providing patients with access to their health information [40]. The increased desire for better patient engagement and more efficient patient-provider communication has pushed the use of patient-facing technology and consumer e-health solutions to empower patients with at-home rehabilitation [40]. Patient-facing tools enable patients to better control their health by allowing them to access their health information, monitor their health status, and manage and follow their treatment at home while recovering [11, 40]. These patient-facing tools can visually display recovery data to enhance patient engagement and patient-provider communication. For example, a wearable sensory display called PTViz was designed to visualize knee rehabilitation for at-home physical therapy for patients recovering from surgery, providing immediate feedback on a range of motion and consequently increasing bodily awareness [2]. Visualizing at-home recovery was also found helpful in addressing the lack of patient engagement in Vestibular Rehabilitation Therapy. Salisbury et al. [42] designed a platform that provides patients with real-time guidance and feedback on therapeutic exercises and allows physical therapists to remotely monitor exercise adherence and performance.

Because of difficulties for outpatients to visit rehabilitation centers for stroke treatment, at-home technological tools have been used to promote patient participation in stroke recovery. For example, MusT, an IoT platform device, was designed to track muscle contraction in the upper limbs of patients recovering from a stroke and send the results to physicians or caregivers to monitor patients' progress and keep them encouraged [55]. Similarly, Ploderer et al. [34] developed a wearable, sensor-equipped prototype that measures arm movements and displays them on dashboards to monitor and visualize patient progress in stroke rehabilitation. Subsequently, researchers broadened this study by developing ArmSleeve [35], a wearable device with interactive dashboards that illustrate how the arm is undergoing robotic treatment, allowing the rehabilitation plan to be adjusted accordingly. Dashboards, which demonstrate how progress was achieved via wearable technology, benefited both therapists and patients. Furthermore, a home-based rehabilitation application, SMART, was developed to record the performance of

daily tasks and rehabilitation exercises for patients recovering from a stroke at home [59]. SMART allows therapists to track patients' progress and provide guidance.

The information gathered from these rehabilitation solutions is important for patients' health monitoring and rehabilitation at home. It can provide significant assistance to healthcare providers and patients, such as feedback on the progress of the therapy program, decision-making, and forecasting future treatment plans.

### 2.2.2 Visualizing recovery in clinic

Prior studies have demonstrated that patients are eager to assist in managing their health while staying in clinics [52]; however, they face barriers to accessing, consuming, sharing, and managing their information [52]. Inpatient hospital settings present unique challenges for patients and caregivers attempting to access, manage, and comprehend information concerning their treatment [10]. Patients in the hospital want to monitor their health but often lack the tools to gather, track, and interpret all their vital data [29]. A number of technological and visualization tools representing recovery progress have been designed to increase patient-provider communication and patient engagement within rehabilitation care in the clinical setting. For example, AnatOnMe [30] is a projection-based handheld device designed to facilitate in-clinic doctor-patient medical information exchange regarding physical therapy. AnatOnMe increased patient engagement in rehabilitation and understanding of medical information. Li et al. [23] also used visualizations representing electromyography biofeedback during physical therapy sessions for patients with acute spinal cord injury, which helped increase muscle use and engagement during therapy.

However, designing and developing patient-facing visualizations of stroke recovery progress in clinical practice has not been thoroughly studied. One factor for the scarcity of patient-facing visualizations of stroke recovery in the inpatient hospital environment might be the notion that patients are already being cared for and do not need further assistance. Although they may be physically looked after, the slow and gradual nature of stroke therapy can make it mentally difficult for stroke survivors to perceive improvement and might lead to dissatisfaction or a lack of enthusiasm toward goal-directed activities. Visualizing and presenting patients with an overview of their progress could significantly enhance their participation and motivation in their treatment. Thus, we aim to address the gap in the literature by taking the first steps to design and develop interactive data visualizations representing patients' overall recovery progress after a stroke while staying in inpatient rehabilitation centers.

## 3 METHODS

We took an iterative user-centred design [45] approach with the involvement of healthcare providers at a local hospital to investigate how to design interactive data visualizations that can best display a patient's stroke recovery progress. First, we conducted semi-structured interviews with healthcare providers experienced in stroke recovery to better understand how patient health outcomes are assessed, how healthcare providers review patient progress over time, and how they communicate rehabilitation progress with patients (Section 3.1). We leveraged the knowledge from the interviews to propose potential visualization designs representing a stroke recovery progress (Section 3.2). Last, we conducted a second round of interviews with healthcare providers seeking their reflection on our suggested visualization designs and asked how they envision using these visualizations in their practice (Section 3.3).

### 3.1 Interviews with Healthcare Providers

*Recruitment:* The managers and physician leads of the stroke rehabilitation unit in the local hospital invited healthcare providers to participate in the study by sharing a research summary form during

their departmental meeting. Additionally, the hospital's communication team placed recruitment posters around the hospital. Interested participants reached out through the e-mail provided, and snowball sampling was used. Through these recruitment methods, we were able to identify healthcare providers whose primary responsibility is performing the rehabilitation of stroke within the inpatient rehabilitation unit. At the end of each interview, we asked the healthcare providers if they knew of other healthcare providers that would be interested in participating in the study. Additionally, we asked the healthcare providers if they would be willing to return for a second interview to discuss the preliminary designs.

*Participants and interview process:* In total, we recruited 4 healthcare providers from a local hospital that have at least one year of experience with stroke recovery within the inpatient stroke rehabilitation unit: a physiatrist (PH) with 30 years of experience, A physiotherapist (PT) with 11 years of experience, a speech-language pathologist (SLP) with 17 years of experience, and an occupational therapist (OT) with 15 years of experience. Interviews were conducted over the phone or online over MS Teams or Zoom. Interviews lasted 30 minutes to 1 hour, depending on the participant's availability and willingness to share. Interviews were audio recorded and later transcribed. Healthcare providers were not allowed to be monetarily compensated for their time per the hospital's review board, which approved all study procedures. The duration of the data collection process was from July to October 2022. Participants agreed to complete an audio-recorded interview after providing oral informed consent.

The interview questions covered 3 main topics: assessing patient health outcomes, reviewing patient progress, and communicating rehabilitation progress to patients. Some example questions include:

- Can you walk me through the steps of your treatment protocol?
- What are the main health outcomes that you assess, and what questionnaires or tests do you use for them?
- How often do you use health assessments?
- To what extent are the patients aware of their progress results, and how is it communicated to them?

*Why a mix of healthcare providers?* Patients in the stroke unit are cared for by a team of healthcare providers with different specialties. Thus, we interviewed healthcare providers with various specialties to gain a complete picture of how each assesses stroke rehabilitation progress.

*Why a low number of healthcare providers?* Recruiting healthcare providers willing to participate in research studies is challenging since they often have a busy schedule or may be skeptical of the value of new technology research [37]. Additionally, our city has only one inpatient stroke unit with 10-12 active healthcare providers offering rehabilitation care for patients after a stroke. We approached all healthcare providers in this clinic and interviewed at least one healthcare provider from each specialty. We have a low number of participants; however, these are specialists with at least 10 years of experience with inpatient stroke rehabilitation; therefore, we consider the number of participants to be sufficient for this study.

*Why didn't we interview patients?* Patients are not often given access to their complete set of health assessment results throughout their stay, and they are not familiar with the necessary health assessments. Thus, we considered healthcare providers as best-suited for answering questions regarding how rehabilitation is measured and communicated to patients.

*Data Analysis:* We reviewed the interview transcripts using an iterative inductive thematic analysis technique for this study [6, 46]. The transcriptions were analyzed individually by two researchers, who then convened to discuss similarities and differences in themes. We coded the data without trying to fit it into a preexisting coding frame or analytic preconceptions. However, as the interviews were semi-structured and covered a relatively narrow range of topics, the

codes typically ended up being organized around the responses to individual questions or categories of similar questions. Then, We classified a set of design requirements that a visualization tool should support to visually represent stroke recovery. Researchers discussed disagreements during the coding process and reached an agreement on case-by-case instances of uncertainty in coding (See Appendix B for the full codebook).

### 3.2 Designing Data Visualization

We sketched various visualization design alternatives to meet the design requirements derived from our interview analysis. The design process began with paper prototypes incorporating design requirements followed by medium-fidelity wireframes produced in Figma. All of the visualizations were designed by one team member and were reviewed by the group. Then, we presented our designs to healthcare providers for feedback.

### 3.3 Evaluating designs with healthcare providers

*Participants and interview process:* To finish our design cycle, we presented our visualization designs to 3 healthcare providers who were part of the first round of interviews and sought feedback from them. Each session lasted between 30 minutes to 1 hour and was video recorded and later transcribed. In this session, we shared the visualization designs with the healthcare providers and observed their reactions while walking through and discussing the ideas aloud.

*Data Analysis:* Similarly to the first round of interviews, we reviewed the interview transcripts using an iterative inductive thematic analysis technique for this study [6, 46]. Transcriptions were analyzed individually by two researchers who then convened to discuss similarities and differences in challenges and pain points observed in the current designs. (See Appendix B for the full codebook). We changed the medium-fidelity prototype designs to address the challenges observed by healthcare providers. All visualization changes were made by one team member and were reviewed as a group.

## 4 RESULTS

To be able to identify requirements to design visualization representing recovery in patients undergoing rehabilitation stroke programs, we interviewed 4 healthcare providers. Our analysis of the interviews revealed 6 themes (T1-T6) that we discussed (Section 4.1). From the themes identified, we defined 5 requirements (DR1-DR5) to design data visualizations representing stroke recovery (Section 4.2). Based on the design requirements, we designed sketches using pen and paper and Figma software (Sections 4.3). We presented our designs to 3 of the healthcare providers who initially participated in the study, gathered their feedback, and then applied their recommendations to our designs (Section 4.4).

### 4.1 Healthcare Provider Interview Results

*T1: Mediums to communicate health progress to patients*

Healthcare providers mentioned ways in which they communicate rehabilitation results and progress to patients. Healthcare providers use verbal communication to discuss rehabilitation progress with patients. Additionally, they share handwritten notes with patients to show their progress. Lastly, patient progress is communicated to patients through communication whiteboards installed in every hospital room. They document patient information such as level of mobility, transfer abilities and discharge dates on these boards.

*T2: Types of content communicated to patients*

Healthcare providers communicate various types of information with patients, including baseline tests, rehabilitation goals, health status scores, discharge summaries, and comparisons between admission and discharge health assessments. The SLP discusses what they typically communicate to patients *"I always have a folder... for all my patients, and I put all the exercises in that. So that the first page of the initial assessment is always there. So then, like, they*

*(patients) can go back to it and review it. And then, at the end, when I do the assessment, I summarize all the information again for the patients. And then I put all those two pages together, and I would say, Okay, now you see and compare. . . so then they can see how much progress they made."*

### T3: Information in a patient's weekly progress report

Healthcare providers mentioned that they assess patients' overall progress over time by discussing aspects of patients' recovery with other healthcare providers in a weekly meeting. This meeting discusses patients' rehabilitation goals as well as their level of function at admission and their progress over time. The PH will discuss any of the patients' medical conditions that may interfere with rehabilitation. The PT discusses the patients' physical functions like mobility, lower-limb function, upper-limb function, transfer, walking ability, and balance. The SLP discusses the patients' cognitive, communication, language, and swallowing abilities. The OT discusses the patients' level of functioning in activities of daily living, cognitive-perceptual, and visual-perceptual functions.

Healthcare providers use similar categories from the Functional Independence Measure (FIM) health assessment as a guide in the team clinical rounds using a scale from 1-7 is to discuss patients' independence in each area of rehabilitation. The PH states *"we do have a one page sheet, which we use to summarize all of the activities of the patient in our rounds, which has basically, the level of function of each patient using, in particular, one weighting system for disability, called the FIM. That stands for the Functional Independence measure. It's a one to seven rating system, which grades the performance of patients in various activities."* In the weekly progress meetings, healthcare providers also discuss the patients' discharge planning, including discharge date, discharge destination and family counselling, as well as follow-up plans, including referrals to outpatient services and healthcare provider follow-up sessions. Each week, the healthcare providers update the patients' functional results and discharge plans, upload them in the EMR software, and communicate with the results to patients. Though it is encouraged for healthcare providers to engage patients by sharing patients' results, healthcare providers mentioned that patients do not typically see all of their health assessment results due to lack of time.

### T4: Attempts to increase patient engagement

Healthcare providers believe that patients should be involved in their rehabilitation journey. Thus, they try to engage patients in their care and focus on increasing patients' knowledge about their health status. To keep patients informed about their health progress, healthcare providers communicate patients' test scores to them. Oftentimes, they conduct assessments in front of the patient so that the patient better understands the results. Test results can be complex and use medical jargon, so providers ensure to use simple language to increase comprehension of the results that they communicate to patients. The PT states *"The tests you're using are complex, so I use simple language for the patients, then they understand and use it as a tool to encourage them to participate."*

Healthcare providers use positive reinforcement, open communication, and refer back to patient goals to encourage patients as they progress in their rehabilitation. The OT states *"I always use open communication. At the end of the session, we usually provide some positive feedback, like, that was really good work today, strong work, I'm happy with the improvements that I've seen in this and this and this. And then I also refer back to the goals that we had established during admission. I would refer back to them and say, you're getting very close to being independent with your self-care, which is what is your goal".*

Additionally, a PT who leads a Virtual Reality (VR) unit in the hospital stated that VR games had made a positive impact on engaging patients in their physical rehabilitation. The VR games provide feedback to the patient on how they're doing and a final summary of their performance in the game. The PT elaborates *"If you have a game that is very straightforward, I've seen ladies in their 80s and 90s in love with going down a virtual ski hill. I love that. And they love it. And they want to know everything about it. And they want to know how they did it. So yeah, definitely really, really engaged."*

### T5: Issues with technology to store and access patient health data

Healthcare providers raised concerns about the current technology used to store and access patient health data. Accessing patient health data from other healthcare providers was reported as challenging if the results were not properly submitted or uploaded in unusual locations. The PH elaborates on the organization of patient health data being incohesive; that EMRs are kept in several locations and are frequently fragmented, PH states *"This stuff [patient health data] is buried in the [EMR software] and it sucks. People always complain that it's very hard to find the level of function and care."* The current EMR software does not have an overview of patient health data, so healthcare providers do not have access to an integrated overview of patient rehabilitation progress. Additionally, healthcare providers think it would be beneficial to have a screen in every room to have all patient information readily accessible.

Healthcare providers expressed a desire for a standard template for summarising patient health progress. The SLP states *"Actually, I wish I had one [template], but it depends on the patient, you know, not all patients are the same. So then, like, there's no one template that I can use, basically."* The uniqueness of each patient's rehabilitation journey makes it difficult to create an interchangeable template for healthcare providers to summarize patient health progress, and personalizing each report is necessary.

Currently, the hospital does not provide exercise tracking devices for patients. The PT emphasizes the importance of patients tracking their cardiovascular health and exercise. To accomplish this, the PT recommends that each patient wear a Fitbit or other exercise tracking device to monitor their heart rate and exercise.

### T6: Inpatient treatment protocol and therapy

Generally, the healthcare provider's protocol begins by conducting a baseline admission assessment to determine the patient's health status upon arrival to the inpatient unit. A team of healthcare providers assesses specific domains of the patient's health status, including the level of cognition, swallowing abilities, language and communication abilities, physical abilities, and medical situations. Rehabilitation goals and treatment plans are then determined based on the level of care the patient needs, the current health challenges they face, and the severity of the patient's condition. In this section, we explain each healthcare provider's treatment protocol to care for patients staying in the inpatient stroke unit of the hospital and the health assessments they use to determine the patient's health status and rehabilitation progress. From our interviews with healthcare providers, we gathered a list of health assessments they use to assess patients' health status and rehabilitation progress (Table 1).

The PH's treatment protocol includes assessing the patient's health status and comorbidities that could be interfering with the patient's rehabilitation, such as hypertension, diabetes, lipids, fever, swelling, and complex regional pain syndrome. PH typically sees patients on a daily basis and uses the SOAP (Subjective, Objective, Assessment and Plan) method, a standardized worldwide method for documenting patients' medical notes in their charts.

The PT treatment protocol includes assessing patients' ability to get in and out of bed, stand and walk. The PT typically finishes admission assessment within the first three days of arrival, which involves inquiring about the patient's social and home environment, mobility, occupation, hobbies, and assessing the ability to perform daily activities. A treatment plan is made depending on the severity of the patients' impairments and typically involves 5 days a week of physiotherapy to work on regaining movement and relearning everyday activities. When applicable, the PT uses assistive devices such as stationary bikes and treadmills to help improve the patient's cardiovascular health.

| Health Assessments | Lay Term | Function |
|---|---|---|
| Assessment of Language-Related Functional Activities (ALFA) [1] | Language-Related Activities | cognitive-linguistic problems related to functional activities |
| Berg Balance Scale (berg) [25] | Balance test | balance levels |
| Boston Naming test [21] | Image Naming test | confrontational word retrieval |
| Box and Blocks test [17] | Block Box and Blocks test | manual dexterity |
| Chedoke McMaster Stroke Assessment [41] | Motor Recovery test | physical impairment and activity |
| Cognitive-Linguistic Quick test (CLQT) [32] | Cognitive-Linguistic test | cognitive-linguistic problems |
| Gait speed test [47] | Walking speed test | gait speed |
| Grip Strength test [9] | Grip Strength test | muscular strength |
| International Dysphagia Diet Standardisation Initiative (IDDSI) [3] | Swallowing test | swallowing ability |
| Montreal Cognitive Assessment (MoCA) [26] | Cognitive test | cognitive impairment |
| Motor Free Visual-Perceptual test [58] | Visual perceptual test | visual perception independent of motor ability |
| Ross Information Processing Assessment-2 [49] | Information Processing test | cognitive-linguistic deficits |
| Star Cancellation test [24] | Star Cancellation test | unilateral spatial neglect |
| Trail Making test [28] | Trail Making test | executive function |
| Western Aphasia Battery (WAB) [4] | Language Test | linguistic skills affected by aphasia |
| 2-Minute Walk test [27] | 2-minute Walk test | endurance and gait speed |
| 9 Peg Hole test [18] | 9 Peg Hole test | finger dexterity |

Table 1: Health assessments used to measure recovery after a stroke, the lay term used in our data visualization, and the function of the health assessments.

The SLP treatment protocol includes assessing patients' language, communication, and swallowing abilities. The SLP assesses for symptoms of aphasia which is a language disorder caused by damage in the brain after a stroke that affects the patient's language expression and comprehension. The SLP also assesses for symptoms of dysarthria which are speech problems often resulting from weak or paralyzed speech muscles. Additionally, the SLP assesses cognitive-linguistic disorders where a patient's attention, concentration and problem-solving may be impacted by a stroke and affects their ability to communicate. Lastly, the SLP assesses the patient's ability to swallow food and drinks. A treatment plan is made depending on the severity of the patient's impairments and typically involves 2-5 speech-language therapy sessions a week. Treatment can involve communication devices, speaking activities, exercises for developing speech muscles, and implementing coping strategies.

The OT treatment protocol includes assessing patients' capacity to perform tasks and comparing it with their pre-stroke self-reported abilities. OT focuses on improving patients' functional capacity in their activities of daily living (personal hygiene or grooming, dressing, toileting, transferring, eating) and instrumental activity of daily living (managing finances, medications, food preparation, house-keeping, laundry). A treatment plan is made depending on the severity of the patient's impairment and typically involves 3-5 occupational therapy sessions per week to improve the patients' motor control and function in the stroke-affected limbs. Additionally, the OT will help the patient make strategies to manage cognitive, perceptual and behavioural changes after a stroke, and prepare the home and work environment for the patient's return.

## 4.2 Design Requirements

From the healthcare provider interview data analysis and themes identified, we extracted 5 design requirements (DR1-DR5) to design data visualization representing recovery progress after a stroke.

- *DR1: Visualize patients' level of independence in daily activities and goals.* This design requirement is drawn from healthcare providers' interview analysis results presented in T3. When asked about visualizing results from the weekly rounds, healthcare providers agreed that it would be beneficial since they use it frequently as a way to discuss the patient's status and rehabilitation progress. The PH states *"I think that [visualizing weekly rounds] will be amazing. Because then we could show it to people and to ourselves. The rounds, they'll become faster and faster."* Since weekly rounds are based on the FIM categories and independence scale, the FIM will be used as a guide for data visualization.

- *DR2: Visualize and categorize patients' health assessments in their associated health domains.* This design requirement is drawn from healthcare providers' interview analysis results presented in T2, T4, and T6. Healthcare providers believe that showing patients their health assessments and exercise results would be beneficial in engaging patients in their rehabilitation. The PT states *"I think they're all [results] important because they give information on the patient on, you know, their rehab, and where they are at some point and where they can be later."* Additionally, the PT mentioned that measuring patients' activity and cardiovascular health could have positive outcomes. To ensure that these results are presented clearly to patients, each health assessment should be placed in their associated health domain, including cognition and perception, language, swallowing, upper-body, lower-body, total motor recovery, and exercise.

- *DR3: Display patients' progress from admission to discharge and display an overview of their progress.* This design requirement is drawn from healthcare providers' interview analysis results presented in T2, T4, and T5. It was noted to be beneficial to communicate patients' progress from admission to discharge for patients to observe how they've been able to progress and achieve their goals over time. The OT gives an example *"Something written down that compares admission and discharge. For example, minimum assistance, moderate assistance, maximum assistance, so at admission self-care, maximum assistance, for upper body dependent for lower, and then on discharge, minimum assistance for the upper and lower body now, showering."* Additionally, as mentioned by healthcare providers, designing the visualization to have an integrated overview of patients' rehabilitation results is necessary to ease the burden of looking for data in a fragmented EMR system.

- *DR4: Using simple language.* This design requirement is drawn from healthcare providers' interview analysis results presented in T2, and T4. Healthcare providers mentioned that

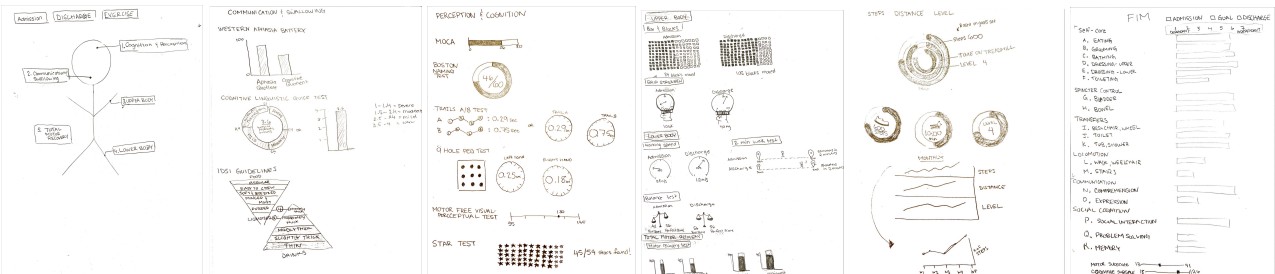

Figure 1: Preliminary sketches displaying rehabilitation progress for balance, cognition, upper/lower limb, mobility, motor recovery and exercise.

the health assessments used can be complicated for patients to understand. To overcome the challenge of communicating the results of complex health assessments, they use simple language to ensure patients are able to comprehend and get engaged. The SLP gives examples *". . . auditory comprehension, for example, I would say 'you're listening'. If it says, 'verbal expression', I would say 'finding the vocabulary' instead of, say, 'written language', I would just say 'writing', instead of saying 'executive content', I would say like 'problem solving or decision-making'."*

- *DR5: Using simple visualization designs charts that are cognitively accessible.* This design requirement is drawn from healthcare providers' interview analysis results presented in T4. The data visualization must address the different levels of comprehension and cognition in patients by using simple-to-understand visualizations. The OT states *"we were dealing with clients who cognitively and perceptually may not be able to manage the type of information we're giving them. So it [visualization] would have to be client-centred as well."*

## 4.3 Preliminary sketches and visualizations

To design the patient health progress visualizations, we considered the requirements (DR1-DR5) identified from our interviews and followed the visualization design guidelines established in the literature. Our preliminary sketches display individual health assessments, health domains, and an overview of patient health assessments laid out in Figure 1. We then designed a medium-fidelity prototype in Figma software to show healthcare providers for evaluation.

To meet DR1, we created a weekly objectives pane that displays the patients' goals and level of function (Figure 2 - A). Each function is represented by a flower that grows in height from level 1 (dependent) to level 7 (independent), similar to the FIM assessment. The grey flower represents the goals; patients can observe their progress towards the goals from admission to discharge and each week in between by scrolling through the time bar representing weeks and watching the flowers grow.

To fulfill DR2, we designed a landing page (Figure 6), where patients are greeted with an overview of all the categories of rehabilitation which are signposted to their relevant body parts on an image of a human. All of the patients' health assessments can be viewed by selecting a button with the associated category of rehabilitation.

To fulfill DR3, we designed an overview of patients' progress between admission and discharge with line graphs that either display the incline or decline in their rehabilitation [12] (Figure 3 - A). From this overview, patients can select a health assessment, which will display a side-by-side comparison of admission and discharge assessment results (Figure 4 - A).

To fulfill DR4, we used lay language to label and described the results of each health assessment for patients (See Table 1). This ensures that the information presented is better understood by the patient. Additionally, in the description of the health assessment,

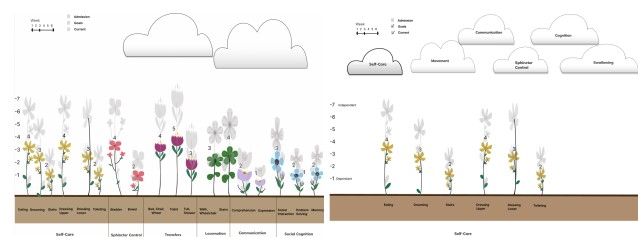

Figure 2: Weekly objectives pane that displays the patients' goals and level of function. left (A) displays all goals with grey flowers and the current level of functioning with coloured flowers, (B) final iteration of the objectives pane with categories of goals placed in clouds to minimize cognitive overload and display progressive disclosure.

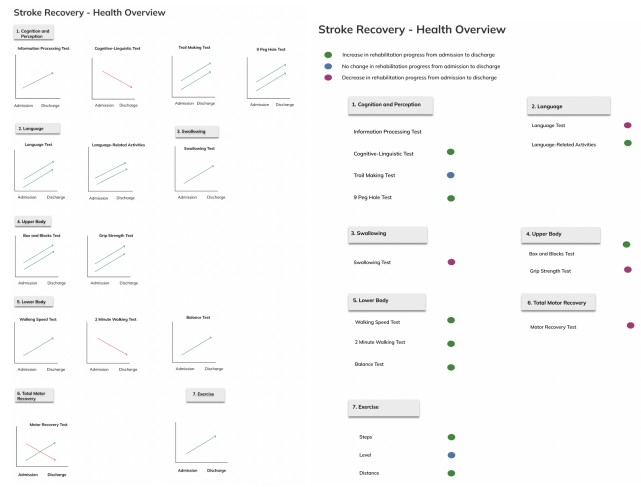

Figure 3: left (A) display of patients' progress overview from admission to discharge represented with line charts, right (B) final design iteration of the overview of the patients' progress overview from admission to discharge represented with coloured circles.

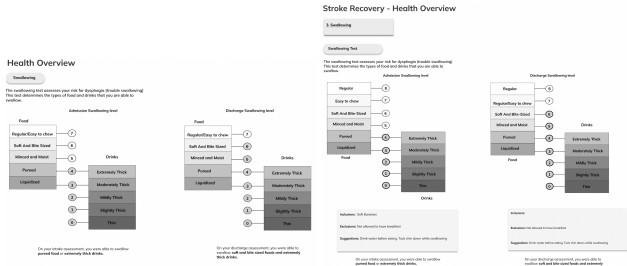

Figure 4: left (A) view of a patient's results from a swallowing assessment done at admission and discharge, right (B) final design iteration of swallowing assessment with added textboxes for personalized food inclusions and exclusions used to help with diet planning.

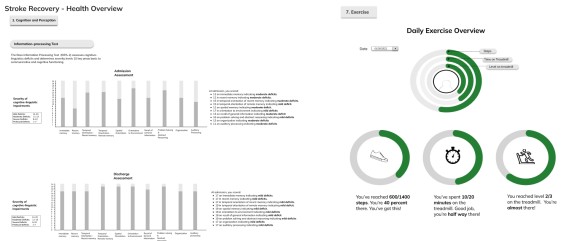

Figure 5: left: (A) display of rehabilitation progress in cognition represented by a bar chart, right (B) display rehabilitation progress in exercise represented by donut charts.

we provided the original name of the test in case the patient decides to share the results with other healthcare providers after discharge. Using a blend of plain language and medical terminology helps patients with varying levels of cognition to easily comprehend their data while providing sufficient medical information for other healthcare professionals to identify the health assessment offered.

To fulfill DR5, we used simple and commonly known graphs and charts. To reduce cognitive overload, patients are presented with an overview alongside details on demand which allows for a progressive display of detailed information [44] (Figure 5).

## 4.4 Evaluation

We presented the visualization designs to three healthcare providers who were interviewed initially and asked for their feedback on the designs. They were instructed to consider a few tasks that they find useful to do while communicating patients' results to them using our visualization tool, such as selecting health evaluations based on body function, selecting the goals panel, selecting the exercise panel, and interpreting results. While we walked through the tasks, we asked healthcare providers to think out loud and tell us about their experiences. After analyzing the data collected during the interviews, we identified 4 themes that we will discuss in this section.

*Clear categorization and labelling of health assessments* Healthcare providers recommended reorganizing a few health assessments into categories that are more related to their health domains. This will assist patients in better understanding trends in their rehabilitation within each domain. Additionally, healthcare providers mention that each category should be signposted to its corresponding body part to assist patients in navigating the visualization and aiding patients in comprehending the nature of the category. For example, a previous iteration of the landing page 6, had 'Communication and Swallowing' as a health category signposted to the mouth of the body, however, we were informed that having two different labels including language signposted to the brain, and swallowing signposted to the throat would be a better representation of health domains.

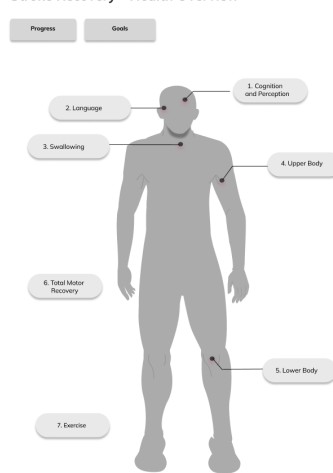

Figure 6: Final design iteration of landing page showing an overview of rehabilitation health domains

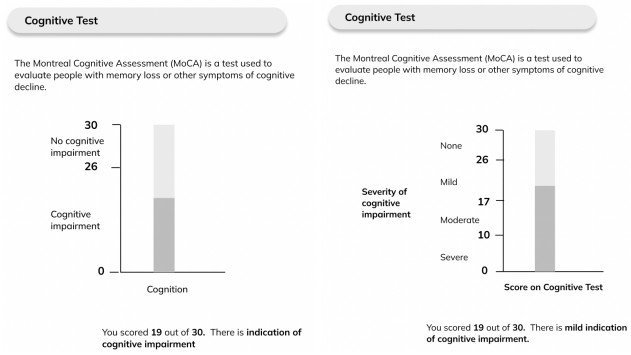

Figure 7: left (A) display of results from a cognitive test at admission, right (B) final design iteration of results from cognitive test at admission with an added severity level of scale.

Last, it was noted that the overview of the patient's progress should display information in a way that is easier to comprehend at first glance and also display when a patient has not progressed in their rehabilitation progress. We integrated these recommendations by replacing the line graphs with circles [12] that have three different colours to represent patients' rehabilitation results either progressing, worsening, or staying the same (Figure 3 - B).

*Display informative scales and results* Healthcare providers mentioned a few ways in which visualization could provide more valuable information to patients. First, adding the severity levels of scales can be informative for patients to better understand the meaning of their results and how they progress in their rehabilitation (Figure 7 - B). The SLP discussed an instance in which their patient's level of impairment in an area of rehabilitation had decreased from severe to mild. In this situation, a scale that only displays 'impairment' and 'no impairment' without specifying the level of impairment misses a chance to appropriately portray the patient's progress. The SLP states *"if I say you were impaired, and now you're impaired, what's the difference? The difference is that it was severe. Now it's mild. So, it's good to tell them how severe it was."*.

Healthcare providers mentioned that rather than visualizing composite scores, the results should display subdomain categories of each health assessment in order to communicate more precise results. Although visualizing a composite score of a health assessment can

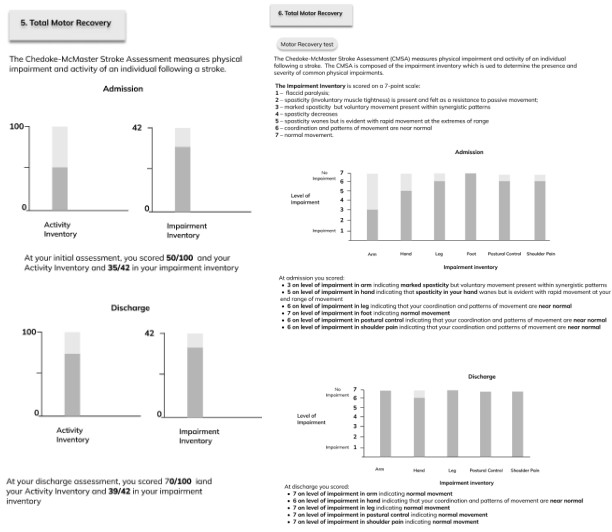

Figure 8: left (A) display of results of composite scores from the total motor recovery test at admission and discharge, right (B) final design iteration of results from the total motor recovery test at admission and discharge with added subdomains.

be useful for minimizing cognitive load, they do not always indicate a particular issue that the patient should be aware of. We integrated this recommendation by displaying subdomains to provide the patient with more valuable and informative results. For example, the display of results from the total motor recovery (Figure 8), now displays results from each subdomain of the impairment inventory rather than one composite score.

Additionally, healthcare providers noted that findings should go beyond the patient's score and should additionally offer an explanation of what the results signify. This better matches how medical professionals often discuss results face to face and will aid the patient in understanding what the scores mean for the course of their rehabilitation. The PH states *"they'll [healthcare providers] put it together in a bigger picture, they won't just write the report as the results. The results will be so integrated to say they improve their self-care; they improve their ability to manage independently, improve their ability to toilet."*

*Personalize the display of information for patients* Healthcare providers believe that the visualization should be personalized for patients' rehabilitation. One way to do this is to allow patients to select which health domains display in the visualization. Allowing the patient to choose which information to display can reduce the cognitive load and allow the patient to easily navigate the results of their rehabilitation. This recommendation was applied to the objectives pane (Figure 2 - B). Instead of displaying all of the patient's goals at once, each category of goals is placed on a cloud. When a cloud is selected, it will display the specific attributes within that category, giving the patient the option to view personalized information.

It was noted that a designated place to add comments could be helpful in communicating patient-specific results. For example, to support patients with diet planning, the swallowing test should include a list of food inclusions and exclusions that the SLP creates for the patient's diet, as well as strategies that the SLP recommends to the patient for swallowing (Figure 4 - B).

Many health assessments include sub-tests that assess various functions in patients' rehabilitation. If the patient performed well on certain sub-tests during the admission assessment, those sub-tests might not be repeated in subsequent assessments. In this instance, when the data visualization displays subsequent results, only sub-tests that have been retested should be displayed in order to provide

the most meaningful and personalized results to patients.

*Personalize healthcare providers' choice of health assessments* Healthcare providers recommended the addition of certain health assessments that are used in their hospital and the deletion of health assessments not used. Each rehabilitation facility may utilize a unique selection of health assessments. Therefore, data visualization must enable healthcare providers to customize the health assessments that will be most beneficial for their rehabilitation centres' protocols.

Our final designs are displayed in the Appendix A.

## 5 DISCUSSION

Effective visualizations of patient health data can encourage patient engagement in their care [2, 34, 40]. Thus, in this study, we created data visualizations representing stroke recovery progress within the inpatient stroke unit to promote patient comprehension and engagement in their rehabilitation. Interviews with healthcare providers revealed the potential benefits of our visualization designs. First, our designs could provide an integrated overview of patient rehabilitation data for patients, as well as inpatient and outpatient healthcare providers. Second, these designs have been shown to have the potential to increase comprehension of rehabilitation progress in patients with cognitive or linguistic deficits. Additionally, our designs could increase communication between patients and healthcare providers as well as between healthcare providers. Finally, we discuss the challenges encountered in conducting studies with healthcare providers and possible future directions.

*Benefits of an integrated overview.* Through interviews with healthcare providers, we found that hospitals' EMRs are difficult to navigate when trying to locate health assessments conducted by other healthcare providers with different specialties [22]. Since health assessments and progress notes are often posted in multiple locations on EMR software, providers feel frustrated for lost time looking for patient information. Our designs address this issue by representing an integrated overview of patients' rehabilitation progress. This overview has the potential to enable providers to examine data from other healthcare providers in a centralized location, saving them time and enhancing communication between healthcare providers.

Patients staying in hospitals want to monitor their health but often lack the means to collect, monitor, and analyze all their data [29]. We discovered that not all health assessments are included in the patient-facing EMR system, leaving patients with limited access to data about their recovery. Although it is unclear which health assessments are uploaded or not, some health assessments are less beneficial to show patients than others. Nevertheless, to effectively engage patients in their care, patients should have the option to overview their health status and treatment information rather than depending on healthcare providers [29]. Additionally, patients typically receive information on their rehabilitation progress through verbal communication, meaning their access to information is contingent upon understanding information on the fly and capacity to recall the information. Patients sometimes receive written reports in different formats from multiple healthcare providers, resulting in the patient having dispersed information. Our proposed visualization designs could provide patients with an organized and cohesive representation that integrates all of their health assessments which could give patients the means to monitor, collect and analyze their rehabilitation data. Additionally, patients have a pressing issue when monitoring their health and care information in a fragmented approach using a combination of paper-based logs, memory, electronic methods, computer, phone applications and digital notes [5]. This approach to monitoring necessitates improved technologies to assist patients and caregivers in tracking health and treatment in and out of hospitals. The integrated overview of patient health progress that our visualization designs provide has the potential to support patients in and out of the hospital by providing an organized summary and reducing the burden on the patient to remember their medi-

cal data. Furthermore, as patients leave the inpatient setting, they could share their rehabilitation outcomes with their new healthcare providers, which will encourage a culture of shared decision-making and cooperation between patients and their providers.

*Visualizations benefit patients with linguistic and cognitive deficits.* We took into consideration the different levels of cognitive and linguistic deficits that patients may face after a stroke so that our visualization designs could address the uniqueness of each patient. To reduce the cognitive burden, we followed the Visual Information-Seeking Mantra: overview first, zoom and filter, then details-on-demand [44]. This gives patients a bird's-eye view of the entire visualization and enables them to progressively reveal more detailed information as needed while filtering out unwanted elements. Additionally, we followed the literature's best practices for designing visualizations for persons with cognitive deficits by avoiding pie charts, encouraging natural metaphors and support for working memory, balancing semantics and simplicity, and employing democratization with axis-aligned encoding [57].

Patients recovering from a stroke may face communication problems, one of which is aphasia, the inability to understand and use language. It can be problematic for patients with aphasia when a healthcare provider verbally communicates or uses written notes, as these patients may have difficulties understanding what is being communicated through language. Visualizing patient health data has the potential to benefit patients with language comprehension deficits by presenting their progress using simple visualizations, which can act as a different medium for patients to understand their results. To further support patients with aphasia, our designs follow a number of recommendations for developing clear documents for people with communication disabilities [39] [7]. These recommendations include using straightforward language, short and simple sentences, bullet points, bolded keywords, images, headings, and sign-posting, which we took into consideration for our designs.

*Challenges and limitations in our study.* Various health assessments are used to test the patients' progress in stroke recovery, and patients do not have access to all of them. During our study, we faced the question of 'How can we identify which health assessments to show patients?' We decided to include the health assessments in our designs that were mentioned to be most commonly used by healthcare providers. We understand and acknowledge that other clinics may use different sets of health assessments. However, we followed the guidelines of the Heart and Stroke Foundation's Canadian Stroke [38] in addition to the recommendations of our healthcare providers. We hope that this is a stepping stone for future research to customizing our visualizations based on individual clinical practices and the patient's characteristics.

We faced a challenge in finding participants for this study. Our city has only one inpatient stroke clinic. Due to their busy schedules, it was challenging to make appointments with healthcare providers. However, we believe that the cumulative expertise of our participants in stroke rehabilitation—more than 60 years—is sufficient for the purposes of our study. We acknowledge the importance of involving the patient in the study. Future work should aim to develop a high-fidelity prototype and evaluate the proposed designs with patients recovering from stroke to gain their feedback and assess the usability of our visualization designs. Future design iterations could customize the designs and display a wider selection of health assessments in order to tailor the visualization designs to a specific hospital that might use other health assessments.

## 6 CONCLUSION

Restoring physical and cognitive abilities after a stroke can be a significant stressor for patients and their families, as well as an economic burden on the healthcare system. Effective visualizations of patient health data can enhance comprehension and encourage patient engagement in care. We presented visualization designs of

stroke recovery progress in an inpatient rehabilitation setting to a team of healthcare providers that each specialize in a certain role within stroke recovery care. Due to the diverse roles of healthcare providers within stroke recovery, it was difficult to build a visualization that fits the needs of all stakeholders. Although there are Canadian standards of health assessments used in stroke recovery, there is no consistency among rehabilitation centres for implementing those guidelines, which might make designing for multiple rehabilitation centres challenging. Our visualization designs have the potential to give patients and healthcare professionals an integrated overview and may lessen the strain of fragmented EMR systems. Providing patients with a comprehensive overview of their stroke recovery progress could improve patients' comprehension of their rehabilitation outcomes, serve as a communication tool between patients and healthcare providers, and be particularly beneficial for patients with cognitive and linguistic deficits, all of which lead to an increase patient engagement in rehabilitation therapy.

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
