# OpenReview forum: "Visualizing recovery progress for patients after a stroke"
_graphicsinterface.org/Graphics_Interface/2023/Conference — Submitted to GI 2023_

### Official Review · Reviewer_KryK · 2023-01-12
**Well-designed research with a major oversight**

**Rating:** 3
**Confidence:** 5

**Review:**

This paper reports on a deep engagement with a set of rehabilitation clinicians to develop visualizations for communicating to their clients progress in the context of stroke recovery. The work begins with interviews to understand the design context. From this themes are developed which are then used to define design requirements. Preliminary designs are then produced which are presented to a subset of the original participants for feedback, which is used to improve the designs. Overall, the study design is very nice and the work is well conducted. I don’t see any problems from a study design or analysis perspective (technically speaking). The paper is well written and easy to follow and understand.

I do however want to challenge the fact that clients were not in a way included in this work even though they are identified as the primary intended user group of the result. This is discussed in the paper in the paragraph “Why didn’t we interview patients?” However, I find this rationale unsatisfactory. The fact that clients are not often given access to their complete data and may not be familiar with the necessary assessments seems a fine rationale for why clinicians were included (and perhaps for why certain questions would be directed only to them), but it does not justify excluding clients, and especially not for excluding them entirely from all stages of the research. Especially in the evaluation stage of the work a lot is assumed about what would or would not work for clients.

In general, the work presents a very medical-model outlook which is out of line with current approaches within the field of HCI, which largely focus on social models of conceptualizing disability that do not position health care providers as experts over the people themselves. The medical influence on this work is apparent even in low level language (e.g. the use of patient vs client). Better engagement with the broader HCI/Accessibility literature (beyond the slice of stroke rehabilitation literature currently included) might help to push back on some of the medical influence.

 At the very least, it is stretch to call this user-centred work.

I see a slim path forward for the research as is, which would require careful word smithing to focus the contribution on clinician perceptions of how to visually communicate progress. This would also require care to ensure the results avoid any assumption of how clients will receive the results.

However, a better path to strengthening the work would be to bring in clients in, at least at the evaluation stage. If working with active clients seems infeasible, perhaps post-recovery clients could be queried and ask to reflect back on their experiences and what their needs were and how these designs may have supported their recovery.

---

### Official Review · Reviewer_hxaH · 2023-01-15
**The paper describes a detailed studies for eliciting requirements for designing interfaces to help stroke patients recover. The authors recruited healthcare professionals for a workshop to understand the utility of visualizations and followed up with them using medium fidelity prototypes. Overall, the paper is well written, motivated and provides useful guidelines for designing systems to help a vulnerable population (stroke patients) and their healthcare providers.**

**Rating:** 8
**Confidence:** 4

**Review:**

The paper describes two studies with healthcare professionals to understand the design requirements for visualization systems to help both stroke patients and healthcare professionals adhere to recovery regimes. The paper is well written and the authors tackle a very important problem. The authors do a good job of surveying the state of the art and position their research as an initial step to utilizing visualizations for stroke recovery.

The authors conducted two iterative user studies. In the first study, the authors interviewed four healthcare providers and drew six themes from the qualitative interview data gathered. The authors then derived five design requirements and used them to design some medium fidelity prototypes for further evaluations. The authors then followed up the initial study with an evaluation of the medium fidelity prototypes with three of the four original healthcare providers. The second study results in four themes for designing visualizations for stroke patients and healthcare providers.

Overall, the authors did a good job of designing the user studies, extracting themes from complex qualitative data and building on feedback from healthcare providers to further improve their designs. The authors provide an important insight into the use of visualizations for stroke recovery by noting that diverse healthcare providers have diverse needs for visualizations and a one-size-fits-all approach may be unsuitable. This is an useful preliminary step to designing visualizations and interactive systems that can support both patients and healthcare providers.

---

### Official Review · Reviewer_YgC6 · 2023-01-17
**Design process paper is not ready for publication**

**Rating:** 4
**Confidence:** 5

**Review:**

The authors of this paper propose to develop visualization tools for helping health care practitioners monitor and understand the recovery patterns of stroke patients over time.   In this paper they discuss a preliminary pilot study with practitioners to determine requirements for visualizations and an interactive design process that resulted in prototype designs. They then had three practitioners comment on these designs presented in medium-fidelity prototypes.  They discuss the feedback with respect to design implications and use it as justification that their visualizations can deliver better outcomes.

I applaud the motivation of this research, but in its current form this paper is much too preliminary to warrant publication.  It is unclear what contribution it actually delivers beyond an introduction to the issues of how practitioners monitor stroke patients' recovery.  It reads very much as a design process report where a series of visualizations are initially developed and then very quickly presented to potential users for comment.  This is a necessary stage in these kinds of projects: the information and insights yielded  serve as the basis for a more rigourous development and study. But in itself it's not clear that there is anything we can really understand about particular potential the visualization designs, the interaction (which is never discussed) or the difference in practices that this might enable, beyond very general insights (for exmaple, visualization can benefit patients with cognitive disability. This is already known!)    I note that while the goal of the general approach was to support patient-facing tools, patients are never interviewed nor part of the feedback process.    One understands the challenge of recruiting such patients, but these limitations need to be discussed.

The writing is pleasant but poorly organized, omitting or obscuring important information. FOr example, rather than providing a Limitations section where the unavoidable shortcomings of the study are discussed (this is always the case), such as the challenge of getting expert users, or the difficulty of assessing long-term benefits in a short interview, the authors spend a good deal of ink justifying it early on (in the first case) or omitting it entirely (in the second).   I would recommend they rad some papers on how to report these sorts of studies.

To sum up, this work has a promising direction, but is as yet still in the formative stages, and the contributions are vague and hard to determine.

---

### Meta-Review · Area_Chair_2SoC · 2023-01-17

**Recommendation:** 4
**Confidence:** 5

**Metareview:**

This paper documents the design of visualizations to support stroke recovery communication through two studies with health care providers.

In terms of strengths, reviewers agreed that the topic is important and that the general direction of the research is worthy and original. There is general consensus that the work is following a solid path that either does or could lead to a meaningful contribution to the field.

One reviewer was quite positive about the work and felt that the results could provide useful guidance to other researchers. Even though the results are quite preliminary that reviewer felt the themes identified were quite useful and could provide guidance to others.

The other two reviewers were more critical, underlining significant flaws in the research. One focuses on the paper's contribution as a design process and considers the work premature for publication. Most specifically, that reviewer felt the results fell short in terms of identifying "the particular potential [of] the visualization designs, the interaction (which is never discussed) or the difference in practices that this might enable, beyond very general insights." This makes it difficult for others to take up the results.

The other reviewer focuses more on the flaws from an accessibility angle, noting as inappropriate the exclusion of people recovering from stroke in the design, and especially evaluation, phases.  This reviewer does leave open the possibility of reframing the work to clarify that it presents _clinicians_ perspectives on how to support recovery progress information and to more clearly emphasize the limitations of not including stroker survivors themselves in the design process (the first reviewer also notes this as a limitation that is not properly discussed but this point is less central to their overall assessment). Reframing would avoid additional data collection, but would require substantial rewriting.

Overall the paper seems premature for publication with two fairly substantial critiques. The reviewers feel each must be addressed prior to publication and both would require substantial additional work (and in at least one case, further research/analysis). As such the recommendation is for rejection.